# A comparative study on folk sports inheritance in China and South Korea: Influencing factors and mediating roles

Xiaobin Jin[1], Yu Hou [2]*

1 Department of Physical Education, Yuncheng Advanced Normal College, Yuncheng, Shanxi, China,
2 Department of Physical Education, Kunsan National University, Gunsan-si, Jeollabuk-do, South Korea

* houyu_sports@qq.com

## Abstract

This study investigates the similarities and differences in the factors influencing folk sports inheritance in China and South Korea, providing a theoretical foundation for fostering bilateral learning and collaboration in this field. The research employs a five-dimensional framework—Inheritance Subject, Inheritance Object, Inheritance Intermediary, Inheritance Environment, and Inheritance Effect—comprising 32 observed variables. Surveys were conducted among 336 Chinese and 331 Korean folk sports participants, organizers, and researchers, with data analyzed using SPSS and AMOS software. The findings indicate a consistent influence of these dimensions on folk sports inheritance in both countries. Specifically, the Inheritance Object, Inheritance Intermediary, and Inheritance Environment positively affect the Inheritance Subject, which, in turn, directly enhances the Inheritance Effect. Furthermore, the Inheritance Subject plays a crucial mediating role within the inheritance system, linking the Inheritance Object, Inheritance Intermediary, and Inheritance Environment while facilitating the realization of the Inheritance Effect. Despite these shared structural influences, notable differences exist in the inheritance processes of folk sports in China and South Korea. South Korea demonstrates strengths in recognizing and cultivating inheritors, establishing institutional frameworks, and developing a sustainable inheritance ecosystem, whereas Chinese folk sports inheritors exhibit superior technical proficiency. This study contributes to the theoretical discourse on folk sports inheritance by highlighting the mediating role of the Inheritance Subject and offering strategic insights for sustaining folk sports traditions in both countries.

## 1 Introduction

In recent years, the processes of modernization, urbanization, and globalization have posed significant threats to the survival of folk sports worldwide, garnering considerable attention from various sectors of society [1,2]. In response to this challenge,

**Data availability statement:** All relevant data are within the manuscript and its Supporting Information files.

**Funding:** The author(s) received no specific funding for this work.

**Competing interests:** The authors have declared that no competing interests exist.

governments and international organizations have implemented a range of initiatives, including policy interventions, legal protections, and cultural advocacy, aimed at preserving and promoting folk sports. Scholars widely agree that folk sports represent a nation's historical and cultural identity, preserve ancestral physical traditions, and constitute important intangible cultural heritage passed down through generations. In addition to their value in promoting physical fitness, education, culture, and tourism, folk sports play a vital role in fostering social cohesion and strengthening rural governance [3–5]. As a result, the preservation and transmission of folk sports have become critical and urgent tasks in contemporary society [6].

Current research on the inheritance of folk sports primarily focuses on logical reasoning studies, including legislative protections for intangible cultural heritage, the integration of folk sports with tourism, case-based analyses of folk sports activities, and the role of event organization in revitalizing these sports [7,8]. However, relying solely on one-dimensional reasoning or case-specific analyses hinders a comprehensive understanding of the mechanisms underlying the inheritance of folk sports. Recently, a small number of scholars have shifted their focus to the multifaceted factors influencing folk sports inheritance and their complex interrelationships. Furthermore, research indicates that systematically delineating the functional mechanisms of these factors is crucial for overcoming inheritance bottlenecks and improving inheritance pathways [9,10].

While this multifactorial perspective represents progress, the existing literature exhibits three significant limitations. First, there is an overreliance on qualitative approaches, with few quantitative analyses based on large-scale empirical data. Second, research remains predominantly confined to single-nation contexts, lacking robust cross-national comparative frameworks that could reveal culturally specific mechanisms. Third, the field lacks a unified analytical model capable of systematically evaluating the multidimensional factors influencing folk sports inheritance and their interrelationships. Addressing these limitations requires a comprehensive investigation of the interactions among these factors to advance the systematic and sustainable inheritance of folk sports, particularly from a holistic perspective on inheritance mechanisms.

China and South Korea, both located in East Asia, share similar geographical conditions and cultural traditions, particularly in terms of the historical lineage of folk sports activities [11]. However, in the contemporary context, the two countries exhibit distinct developmental patterns and differing statuses in the inheritance of folk sports [12]. While China possesses a rich and diverse heritage of folk sports, it faces challenges related to inheritance awareness, the training of inheritors, and the preservation of the inheritance ecosystem [13,14]. In contrast, South Korea excels in the institutional protection of intangible cultural heritage and the systematic recognition of inheritors [15]. This "differentiated development on a shared cultural foundation" underscores deeper disparities between China and South Korea in areas such as folk sports promotion policies, preservation environments, and inheritance mechanisms. A cross-national empirical comparison of these two countries can reveal key

determinants of folk sports inheritance, providing a more comprehensive and globally informed academic perspective on its complex mechanisms.

To further clarify the factor system influencing the inheritance of folk sports in China and South Korea, this study conducts a targeted comparative analysis of specific indicators to provide constructive insights for improving China's Inheritance Environment. Building on existing literature, a novel structural equation model of influencing factors is developed to compare folk sports in both countries across five dimensions: Inheritance Subject, Inheritance Object, Inheritance Intermediary, Inheritance Environment, and the corresponding Inheritance Effect [9,10]. A survey was administered to relevant participants in both countries, and the collected data were analyzed using SPSS software for comparative assessment.

This study makes three key contributions, each addressing a major gap in prior research. First, it addresses the methodological overreliance on qualitative approaches by employing large-scale surveys in China and South Korea and applying structural equation modeling to validate the factor structure of folk sports inheritance. Second, it transcends single-nation perspectives through a cross-national comparative framework that reveals both shared foundations and divergent developments. Third, it addresses the lack of a unified analytical model by proposing and empirically testing a five-dimensional framework—Inheritance Subject, Object, Intermediary, Environment, and Effect—and further highlights the mediating role of the Inheritance Subject. Collectively, these contributions not only clarify key mechanisms but also provide targeted recommendations for China and a replicable framework for international scholarship.

The organization of this paper is as follows: The second section begins with a literature review, presenting the theoretical background of the study, followed by an introduction to the research design. The fourth section presents the research findings, while the fifth section discusses these findings. Finally, the conclusion section offers recommendations for improving the current folk sports inheritance practices in China.

## 2 Literature review

Studies on the influencing factors of folk sports inheritance in China have evolved from fragmented explorations to a more systematic research framework. Zha et al. suggest that government involvement and the rise of modern competitive sports significantly influence folk sports inheritance [3]. Chen emphasizes that cultivating active inheritance awareness, strengthening infrastructure, and prioritizing folk sports education are essential strategies for promoting folk sports [4]. In terms of strengthening folk sports education, Zha and Zhang highlight that developing and utilizing folk sports curriculum resources are effective approaches. In the era of big data, Indah et al. identify the scarcity of digital resources and the shortage of multidisciplinary talent as key constraints to integrating folk sports into contemporary society [2,16]. These studies explore the topic from different perspectives, including governmental policies, public perceptions of inheritance, and the integration of folk sports into education. However, they remain fragmented and limited in their scope. Recently, some scholars have begun conducting systematic investigations into the factors influencing folk sports inheritance, providing a more holistic understanding of the subject. Cui et al. conducted an integrated analysis of the dynamic inheritance factors and identified the Inheritance Subject as the pivotal element [9]. They further determined that government support, social endorsement, cultural environment, project adaptability and innovation, and material conditions function as exogenous latent variables that directly impact the Inheritance Subject. Zhang et al. examined the inheritance of kite culture as a case study, categorizing the influencing factors into four areas: cultural heritage transmission, Inheritance Subject, Inheritance Audience, and social environment [17]. They identified 16 specific indicators for systematic investigation, concluding that policy regulations and project attributes are primary determinants of intangible cultural heritage inheritance. Ma and Wang applied grounded theory's "three-level coding" method to extract 40 initial categories and four principal categories, constructing a theoretical model of folk sports inheritance factors centered around four key dimensions: Inheritance Subject, Inheritance Object, Inheritance Intermediary, and external environment [10]. The 40 influencing factors derived from this model form the structural basis for this study's comparative investigation into folk sports inheritance in China and South Korea.

Scholars generally agree that Korea has developed advanced and well-established practices in organizing folk festivals, protecting the ecological environment, and managing intangible cultural heritage, particularly in the context of folk sports inheritance. Lee found that the celebration of the Gangneung Dano Festival has revitalized the inheritance of folk sports such as wrestling, swinging, archery, and tug-of-war [18]. Han and Jeong observed a direct correlation between changes in production and lifestyle patterns and the weakening of the traditional game inheritance system [19]. Similarly, Ryu argued that geographical factors, labor conditions, and local residents' attachment to their hometowns significantly impact regional cultural heritage transmission [20]. Tao compared Korea's tug-of-war with China's dragon boat racing, concluding that cultural value identity, significant festival timing and space, government protection policies, and the integration of folk sports into school education are key factors influencing folk sports inheritance [21]. Studies by Zheng and Ma et al. highlight that Korea's experience in protecting intangible cultural heritage in sports primarily involves systematic legal frameworks, a well-established mechanism for cultivating inheritors, and a diverse range of festival sports events [15,22].

At a deeper level, the theoretical interpretation of the influencing factors in folk sports inheritance is rooted in a multidisciplinary perspective. From the standpoint of intangible cultural heritage (ICH) studies, the inheritor is regarded as the creator and bearer of folk sports, referred to as the Inheritance Subject [23]. The Inheritance Object denotes the intangible cultural heritage itself, which in this study refers specifically to distinct folk sports programs [24]. Based on the theory of cultural tools [25], the Inheritance Intermediary functions as a bridge between the Inheritance Subject and the Inheritance Object, facilitating their interaction and encompassing elements such as awareness of protection and institutional mechanisms. From an anthropological perspective, the natural and social environment in which a sport exists plays a vital role in the development of ethnic sports; in this study, it is defined as the Inheritance Environment [26]. From a management science perspective, the Inheritance Effect serves as a direct indicator and feedback mechanism for evaluating the effectiveness of cultural transmission, and is an essential component of the inheritance mechanism of traditional ethnic sports culture [27]. Collectively, the five dimensions—Inheritance Subject, Inheritance Object, Inheritance Intermediary, Inheritance Environment, and Inheritance Effect—form a complete and dynamic coupling system that provides a theoretical lens for analyzing the differences in folk sports inheritance pathways between China and South Korea.

In summary, while China faces an urgent need to enhance folk sports inheritance and Korea's institutionalized model offers valuable insights, a direct comparative study remains absent despite its urgency. This gap is particularly critical, as rapid modernization in China accelerates the decline of traditional practices [19,20], whereas Korea's established system of legal protection and festival revitalization provides a mature counterpoint [15,22]. However, existing research fails to deliver cross-national insights due to three limitations: first, an overreliance on qualitative rather than quantitative analyses; second, a focus confined to single-nation contexts without a structured comparative framework; and third, the absence of a unified analytical index system capable of evaluating the complex interplay of factors. To address these gaps, this study employs a large-scale quantitative survey to construct a five-dimensional analytical framework—Inheritance Subject, Object, Intermediary, Environment, and Effect. Specifically, it seeks to address the following core questions:

(1) Does the Inheritance Object influence the Inheritance Subject? To what extent?

(2) Does the Inheritance Intermediary influence the Inheritance Subject? To what extent?

(3) Does the Inheritance Environment influence the Inheritance Subject? To what extent?

(4) Does the Inheritance Subject influence the Inheritance Effect? To what extent?

(5) Are there significant differences in the influencing factors of folk sports inheritance between China and Korea?

(6) Does the Inheritance Subject play a mediating role in the overall inheritance system? What is its mechanism?

It should be noted that, by comprehensively referencing existing research frameworks and revisiting the definition of "Inheritance Object" proposed by folklore scholar Jiang, "folk sports events themselves" have been selected as the "Inheritance Object" to reorganize the related influencing factors [9,10,28]. After logical adjustment and conceptual alignment of the impact indicators across the five major categories, the following research framework diagram for this article has been constructed (as shown in **Fig 1**).

## 3 Methods

### 3.1 Research sample

The research sample in this study consists of four groups: inheritors of folk sports, participants in folk sports, researchers and educators in the field of folk sports, and administrators involved in folk sports. In China, a total of 370 questionnaires were distributed using the Wenjuanxing professional online survey platform. In South Korea, 350 questionnaires were distributed via Google Forms, a platform widely used and familiar to Korean respondents. Two types of invalid responses were identified: (1) questionnaires containing contradictory answers, such as rating an item as "very good" in one section and "very poor" in another, and (2) questionnaires where respondents selected the same answer for all items, such as choosing option 1 throughout. After excluding these invalid responses, 336 valid questionnaires were obtained from China and 331 from South Korea, resulting in a total of 667 valid samples. The sample includes respondents of diverse genders, ages, occupations, and educational backgrounds, thereby providing a representative and broad data foundation for subsequent research and analysis on folk sports participation in both countries.

To enhance transparency, the sampling procedure adopted a stratified approach across four key stakeholder groups—inheritors, participants, educators/researchers, and administrators—within diverse regions of both China and South Korea. This ensured representation from both practice-oriented and policy-oriented communities, consistent with inclusivity principles outlined in the Supporting Information (SX Checklist). Variable inclusion followed a multi-step rationale: indicators were first derived from existing theoretical models, then refined through pilot testing with 120 respondents, during which

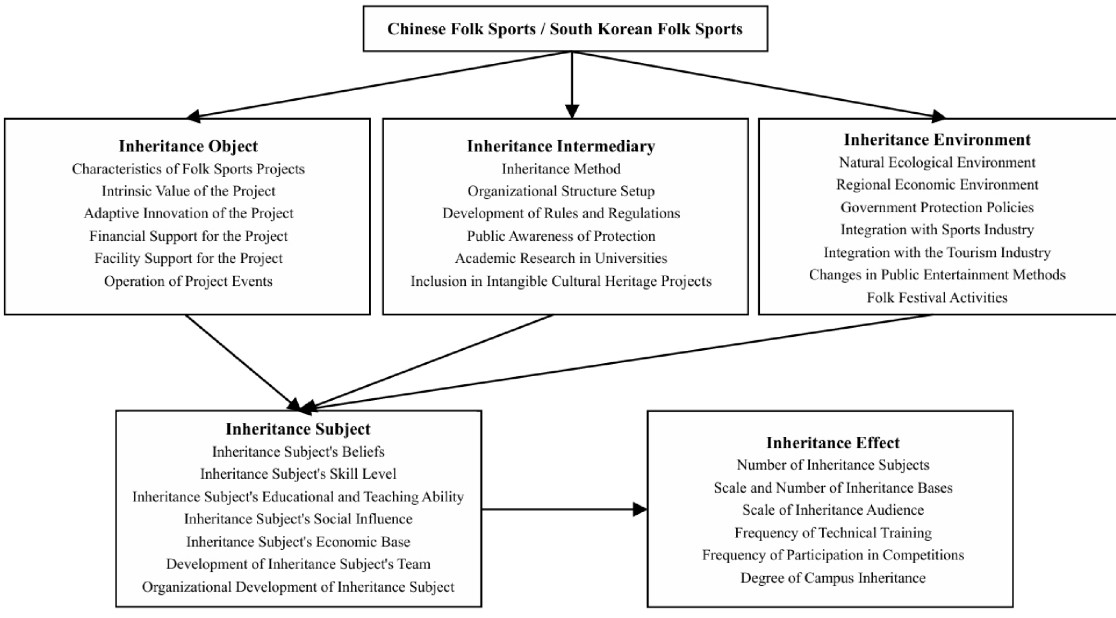

**Fig 1. Research framework of the article.**

items with low factor loadings (<0.50) or conceptual overlap were excluded. All questionnaires were completed in full, resulting in no missing data. Consequently, no imputation was required, and all valid samples were included in the analysis. This process aligns with best practices in quantitative heritage research and strengthens the robustness of subsequent SEM analyses.

### 3.2 Determination of research variables

To thoroughly explore the influencing factors of folk sports inheritance in China and Korea, this study constructs a variable system based on existing literature and related theoretical models [9,10]. The study analyzes five dimensions: Inheritance Subject, Inheritance Object, Inheritance Intermediary, Inheritance Environment, and Inheritance Effect. Among these, the Inheritance Object, Inheritance Intermediary, and Inheritance Environment jointly influence the Inheritance Subject, while the Inheritance Subject directly affects the Inheritance Effect. Following the initial establishment of the variable system, preliminary research was conducted through interviews and surveys targeting individuals who have participated in or organized folk sports activities. A total of 120 questionnaires were distributed, and the collected data were evaluated for reliability, validity, and the rationality of the observed variable settings. Based on the evaluation results, necessary adjustments and modifications were made to the questionnaire, and the core variable system for this study was ultimately established.

Inheritance Subject: Seven observation variables were set, covering concepts such as skill level, educational and teaching level, social influence, economic foundation, team-building, and organizational development.

Inheritance Object: Six observation variables were set, including characteristics of folk sports projects, their inherent value, adaptive innovation, financial support, venue and facility guarantees, and event operation.

Inheritance Intermediary: Six observation variables were set, including transmission methods, organizational structure, rule and system construction, public protection awareness, university-level theoretical research, and intangible cultural heritage projects.

Inheritance Environment: Seven observation variables were set, including the natural ecological environment, regional economic environment, government protection policies, integration with sports undertakings, integration with the tourism industry, changes in public entertainment modes, and folk festival activities.

Inheritance Effect: Six observation variables were set, including the number of inheritors, the scale and number of inheritance bases, the scale of the audience, frequency of technical training, frequency of competition participation, and the degree of inheritance in schools.

### 3.3 Ethical approval and permissions

Ethical approval for this study was granted by the Ethics Committee of Yuncheng Advanced Normal College, China (Approval No.: 2024−0036). The study was conducted in accordance with the principles of the Declaration of Helsinki. All participants provided informed electronic consent before completing the survey. Consent was obtained in written form via the online platforms (Wenjuanxing in China and Google Forms in South Korea), where participants confirmed that they had read and understood the study purpose, the voluntary nature of participation, and the confidentiality of their responses. This study did not involve minors, and no waivers of consent were granted by the ethics committee.

In South Korea, the study was conducted independently by the Korean corresponding author, and additional ethics approval was not required under the local institutional policy. The research adhered to all ethical guidelines established by South Korean institutional norms.

Furthermore, local cultural heritage institutions, community organizers, and folk sports administrators in both countries were informed about the study's objectives, and their informal consent was obtained through discussions regarding the study's design. While formal written consent from community leaders was not required, the research team collaborated with these stakeholders to ensure the study aligned with local cultural practices and ethical considerations.

## 3.4 Survey distribution and evaluation

This study developed two identical questionnaires in Chinese and Korean, utilizing the Chinese professional online survey platform Wenjuanxing and the widely used Google Forms in South Korea for data collection [29]. Respondents who successfully completed the survey received a reward of 10 RMB upon completion. A five-point Likert scale was used to evaluate the questionnaires. Participants were asked to rate each observed variable based on their personal perceptions, where 1 indicated "strongly disagree," 2 indicated "disagree," 3 indicated "neutral," 4 indicated "agree," and 5 indicated "strongly agree".

## 3.5 Data analysis methods

No personally identifiable information was collected during the study. All responses were anonymous, and the survey platforms automatically encoded the data. Only aggregated data were accessed and analyzed using IBM SPSS Statistics 26.0 and IBM SPSS Amos 26.0. The research team ensured that all ethical and data protection guidelines were strictly followed.

After confirming the validity and reliability of the questionnaires, this study will employ two statistical software programs, SPSS 26.0 and AMOS 26.0, for data analysis. SPSS will be primarily used for descriptive statistics, reliability analysis, validity analysis, and correlation analysis, while AMOS will be used to validate the Structural Equation Model (SEM), explore the path relationships between variables, and address the research questions.

## 3.6 Inclusivity in global research

In conducting this comparative study of folk sports inheritance in China and South Korea, several ethical, cultural, and scientific considerations were taken into account to ensure inclusivity and respect for the diverse contexts of both countries. The research aimed to bridge cultural differences while honoring the traditions, values, and norms specific to each region. Ethical approval for the study was granted by the Ethics Committee of Yuncheng Advanced Normal College, China (Approval No.: 2024−0036). The study adhered to the ethical guidelines established in the Declaration of Helsinki, ensuring that all participants provided informed consent electronically before participating. In South Korea, the study was conducted independently by the Korean corresponding author, and additional ethics approval was not required under local institutional policies. All participants were informed of their voluntary involvement, the confidentiality of their responses, and their right to withdraw at any point without consequences.

Given the differing cultural contexts between China and South Korea, the study was designed to be culturally sensitive. The surveys were tailored to the language and cultural nuances of both countries, ensuring that questions were easily understandable and relevant to the respondents in each region. Special care was taken to respect the local traditions of folk sports and the roles of inheritors in each society. Additionally, the study collaborated with local cultural heritage institutions, community organizers, and folk sports administrators to ensure that the research design aligned with local cultural practices and values.

The study incorporated diverse demographic groups, including folk sports inheritors, participants, researchers, educators, and administrators from both countries. This inclusivity allowed for a comprehensive representation of the various stakeholders involved in folk sports inheritance, ensuring that multiple perspectives were captured. The survey responses were anonymized to protect the privacy of participants, and only aggregated data were analyzed. The inclusion of both male and female participants, as well as individuals from various age groups and educational backgrounds, further enhanced the representativeness of the sample.

In order to make this research scientifically robust, the study employed a systematic approach to data collection and analysis. The use of SPSS and AMOS software for statistical analysis allowed for objective comparisons between the two countries, ensuring that the findings were based on sound scientific methodology. The inclusion of multiple

variables—such as Inheritance Subjects, Objects, Intermediaries, Environments, and Effects—enabled a thorough examination of the complex factors influencing folk sports inheritance, while accounting for the unique cultural contexts in each country.

Additional information regarding the ethical, cultural, and scientific considerations specific to inclusivity in global research is included in the Supporting Information (SX Checklist), which provides further details on how these aspects were addressed during the study.

## 4 Results and analysis

### 4.1 Comparative analysis of demographic data of respondents from China and South Korea

Among all respondents, the proportion of males is higher than that of females in both countries, with a notably higher proportion of males in Korea. In terms of age distribution, both countries exhibit similar patterns across age groups, although the proportion of individuals aged 65 and above is slightly higher in China. Regarding the distribution of respondents, Korea has a higher proportion of project participants, while China has a higher proportion of project inheritors. In terms of education, the overall educational level of Korean respondents is higher, with a notably greater proportion holding a bachelor's degree compared to those in China. See Table 1 for details. Overall, the demographic data related to the inheritance of folk sports in both China and Korea reveal both similarities and differences. In both countries, males are more involved in folk sports inheritance than females, which may be linked to the traditional dominance of men in sports activities. Regarding education, the higher educational level of Korean respondents, particularly the significantly greater proportion of individuals holding a bachelor's degree, may be attributed to Korea's developed education system in recent years and the public's strong emphasis on higher education.

### 4.2 Reliability and validity of the measurement tools

This study conducts reliability analysis using Cronbach's Alpha coefficient. According to the relevant literature, a Cronbach's Alpha value greater than 0.7 indicates that the scale's reliability meets the required standards and demonstrates high internal consistency, permitting further analysis. If the value is below 0.7, adjustments to the questionnaire or an increase in sample size are necessary. Initially, SPSS 26.0 is used to perform factor analysis on the variables of the overall sample, eliminating items with factor loadings less than 0.5. The reliability coefficients of latent variables are then

**Table 1. Demographic data of Chinese and South Korean respondents.**

|  |  | China | (%) | South Korea | (%) |
|---|---|---|---|---|---|
| Gender | Male | 185 | 55.1 | 199 | 60.1 |
|  | Female | 151 | 44.9 | 132 | 39.9 |
| Age | Under 25 years old | 27 | 8.0 | 27 | 8.2 |
|  | 26-45 years old | 95 | 28.3 | 96 | 29.0 |
|  | 46-65 years old | 135 | 40.2 | 136 | 41.1 |
|  | Over 65 years old | 79 | 23.5 | 72 | 21.8 |
| Surveyed Subjects | Project Inheritors | 98 | 29.2 | 70 | 21.1 |
|  | Project Participants | 155 | 46.1 | 176 | 53.2 |
|  | Teaching and Research Personnel | 52 | 17.3 | 60 | 18.1 |
|  | Relevant Administrators | 25 | 7.4 | 25 | 7.6 |
| Education Level | Secondary School | 81 | 24.1 | 30 | 9.1 |
|  | Undergraduate | 182 | 54.2 | 219 | 66.2 |
|  | Postgraduate | 73 | 21.7 | 82 | 24.8 |
| Total |  | 336 | 100.0 | 331 | 100.0 |

calculated to assess whether the empirical data of each latent variable meet the internal consistency requirements. The results are presented in Table 2.

Confirmatory Factor Analysis (CFA) was conducted using AMOS 26.0 on the selected indicators to test their validity. When the results align with the variable structure analysis, the model fit indices and standardized factor loadings can be used to assess convergent validity. According to Fornell and Larcker's criteria, three conditions for determining convergent validity are as follows: ① All standardized factor loadings exceed 0.5; ② Composite reliability (CR) is greater than 0.6; ③ Average variance extracted (AVE) is greater than 0.5. This study examines convergent validity through construct reliability (CR) and average variance extracted (AVE). A CR value greater than 0.7 and an AVE value greater than 0.5 are considered to meet the standard. The results, as shown in Table 3, indicate that the factor loadings of each observed variable range from 0.680 to 0.863, all exceeding the recommended threshold of 0.5. The composite reliability values range from 0.8626 to 0.9245, significantly surpassing the standard value of 0.6. The AVE values range from 0.5127 to 0.6364, and the AVE for all other latent variables in each dimension exceeds the recommended value of 0.5. The bolded content in Table 4 shows the square roots of AVE for each dimension, along with the correlation coefficients and the square roots of AVE. For the five latent variables in the Chinese and Korean survey samples, the square roots of their AVE are greater than the correlation coefficients between them and other factors, further confirming that the model exhibits strong discriminant validity, making it suitable for further research. The results show that the standardized loading coefficients of each observed variable on the corresponding latent variable exceed 0.5, indicating that each observed variable can largely explain its latent variable, demonstrating sufficient convergent validity for all variables. In summary, the confirmatory factor analysis indicators in this study meet the required standards, and the overall model fit is adequate.

## 4.3 Empirical results

This study first conducts a model fit accuracy analysis using AMOS 26.0 software, followed by the application of Structural Equation Model (SEM) to test the hypotheses (Table 5). The results of the model fit evaluation indicate that the model demonstrates a good fit and is acceptable. According to the standards of model fit indices, the model's fit indices meet the required criteria. Therefore, the paths of the model were analyzed, and the results of the path analysis are presented in Table 6.

Standardized effect sizes ($\beta$ coefficients) were reported alongside p-values for all tested paths to improve interpretability. Confidence intervals for these coefficients were estimated using bias-corrected bootstrapping with 5,000 resamples. For mediation analyses, standardized indirect effects and their 95% confidence intervals are presented in Table 7. This approach facilitates comparability between China and South Korea and adheres to APA and SEM reporting guidelines. Tables 6 and 7 present effect sizes and significance levels in parallel, allowing simultaneous assessment of statistical and practical significance.

**4.3.1 Analysis of the relationship test results between inheritance object and inheritance subject.** The Inheritance Object significantly influences the Inheritance Subject in both China and South Korea ($\beta$_China = 0.390,

**Table 2. Cronbach's alpha coefficients of the measurement instruments.**

| Latent variable | Number of items | Cronbach's alpha coefficients | |
| --- | --- | --- | --- |
| | | China | South Korea |
| Inheritance Subject | 7 | 0.925 | 0.890 |
| Inheritance Object | 6 | 0.904 | 0.875 |
| Inheritance Intermediary | 6 | 0.888 | 0.882 |
| Inheritance Environment | 7 | 0.909 | 0.894 |
| Inheritance Effect | 6 | 0.862 | 0.875 |

**Table 3. Convergent validity analysis of variables.**

| | | Factor loading | | CR | | AVE | |
|---|---|---|---|---|---|---|---|
| | | China | South Korea | China | South Korea | China | South Korea |
| Inheritance Subject | Inheritance Beliefs | 0.787 | 0.741 | 0.925 | 0.890 | 0.636 | 0.536 |
| | Skill Level | 0.783 | 0.705 | | | | |
| | Teaching Level | 0.755 | 0.717 | | | | |
| | Social Influence | 0.841 | 0.761 | | | | |
| | Economic Foundation | 0.790 | 0.725 | | | | |
| | Team Building | 0.825 | 0.749 | | | | |
| | Organizational Development | 0.800 | 0.723 | | | | |
| Inheritance Object | Project Characteristics | 0.795 | 0.800 | 0.905 | 0.876 | 0.614 | 0.542 |
| | Intrinsic Value | 0.756 | 0.643 | | | | |
| | Project Innovation | 0.761 | 0.666 | | | | |
| | Funding Support | 0.818 | 0.737 | | | | |
| | Facility Support | 0.780 | 0.735 | | | | |
| | Event Operation | 0.789 | 0.819 | | | | |
| Inheritance Intermediary | Inheritance Methods | 0.780 | 0.766 | 0.889 | 0.883 | 0.574 | 0.559 |
| | Institutional Setup | 0.731 | 0.765 | | | | |
| | Regulations and Systems | 0.734 | 0.692 | | | | |
| | Awareness of Protection | 0.683 | 0.683 | | | | |
| | Theoretical Research | 0.764 | 0.716 | | | | |
| | Project Selection | 0.843 | 0.850 | | | | |
| Inheritance Environment | Natural Environment | 0.680 | 0.705 | 0.909 | 0.895 | 0.591 | 0.548 |
| | Regional Economy | 0.705 | 0.729 | | | | |
| | Protection Policies | 0.863 | 0.743 | | | | |
| | Sports Integration | 0.785 | 0.795 | | | | |
| | Tourism Integration | 0.762 | 0.725 | | | | |
| | Entertainment Methods | 0.754 | 0.693 | | | | |
| | Festival Activities | 0.815 | 0.787 | | | | |
| Inheritance Effect | Number of Inheritants | 0.769 | 0.774 | 0.863 | 0.876 | 0.513 | 0.542 |
| | Inheritance Bases | 0.668 | 0.729 | | | | |
| | Audience Size | 0.627 | 0.645 | | | | |
| | Technical Training | 0.714 | 0.719 | | | | |
| | Competition Frequency | 0.726 | 0.729 | | | | |
| | Campus Inheritance | 0.78 | 0.812 | | | | |

$p < .001$; $β\_Korea = 0.161$, $p < .01$). The stronger effect in China suggests that the intrinsic characteristics and cultural value of folk sports projects act as primary motivators for inheritor engagement. These factors foster a sense of recognition and belonging, thereby enhancing willingness to participate. In Korea, the weaker yet significant effect indicates reduced reliance on the Inheritance Objects themselves, which may reflect the country's more developed institutional framework, including training programs, certification systems, and legal protections that provide alternative sources of motivation. Overall, these findings reveal two contrasting inheritance mechanisms: the Chinese model is more culturally and identity-driven, whereas the Korean model depends more heavily on institutional reinforcement.

**4.3.2 Analysis of the relationship test results between inheritance intermediary and inheritance subject.** The Inheritance Intermediary significantly influences the Inheritance Subject in both China and South Korea ($β\_China = 0.181$, $p < .001$; $β\_Korea = 0.352$, $p < .001$). In China, this effect reflects contributions from organizational structures and

**Table 4. Correlation coefficients and discriminant validity analysis.**

| | Inheritance object | Inheritance intermediary | Inheritance environment | Inheritance effect | Inheritance subject |
|---|---|---|---|---|---|
| **China** | | | | | |
| Inheritance Object | **0.798** | | | | |
| Inheritance Intermediary | 0.532 | **0.783** | | | |
| Inheritance Environment | 0.519 | 0.491 | **0.757** | | |
| Inheritance Effect | 0.348 | 0.498 | 0.458 | **0.768** | |
| Inheritance Subject | 0.661 | 0.552 | 0.559 | 0.266 | **0.716** |
| **South Korea** | | | | | |
| Inheritance Object | **0.732** | | | | |
| Inheritance Intermediary | 0.452 | **0.736** | | | |
| Inheritance Environment | 0.523 | 0.547 | **0.747** | | |
| Inheritance Effect | 0.466 | 0.531 | 0.502 | **0.740** | |
| Inheritance Subject | 0.499 | 0.587 | 0.614 | 0.398 | **0.736** |

**Table 5. Model fit assessment (Model without MI adjustments).**

| Fit Index | χ²/df | CFI | NFI | GFI | AGFI | IFI | RMSEA |
|---|---|---|---|---|---|---|---|
| China | 1.969 | 0.934 | 0.875 | 0.864 | 0.843 | 0.934 | 0.054 |
| South Korea | 1.737 | 0.940 | 0.871 | 0.869 | 0.848 | 0.941 | 0.047 |
| Acceptable Value | < 3 | >0.9 | >0.9 | >0.8 | >0.8 | >0.9 | <0.08 |

**Table 6. The relationships between variables and their statistical tests.**

| | Path Relationship | China | | | | South Korea | | | |
|---|---|---|---|---|---|---|---|---|---|
| | | Standardized Path Coefficient | S.E. | C.R. | Result | Standardized Path Coefficient | S.E. | C.R. | Result |
| 1 | Inheritance Object→Inheritance Subject | 0.390 | 0.053 | 7.356*** | + | 0.161 | 0.046 | 2.777** | + |
| 2 | Inheritance Intermediary→Inheritance Subject | 0.181 | 0.050 | 3.596*** | + | 0.352 | 0.05 | 5.542*** | + |
| 3 | Inheritance Environment→Inheritance Subject | 0.307 | 0.070 | 4.382*** | + | 0.404 | 0.069 | 5.787*** | + |
| 4 | Inheritance Subject→Inheritance Effect | 0.368 | 0.066 | 5.588*** | + | 0.515 | 0.082 | 7.913*** | + |

Note: **$p < .01$, *** $p < .001$

**Table 7. Mediating effect test results of the inheritance subject.**

| Mediating Effect Path | Significance (two-tailed test) | | 95% of Confidence Interval | | | |
|---|---|---|---|---|---|---|
| | | | Lower Limit | | Upper Limit | |
| | China | South Korea | China | South Korea | China | South Korea |
| Inheritance Object→Inheritance Subject→Inheritance Effect | 0.160 | 0.083 | 0.100 | 0.017 | 0.221 | 0.157 |
| Inheritance Intermediary→Inheritance Subject→Inheritance Effect | 0.070 | 0.181 | 0.026 | 0.110 | 0.123 | 0.256 |
| Inheritance Environment→Inheritance Subject→Inheritance Effect | 0.085 | 0.208 | 0.039 | 0.131 | 0.141 | 0.289 |

theoretical research, which have enhanced inheritors' educational capacities and facilitated organizational development, but are still at a relatively early stage. By contrast, the stronger effect in South Korea underscores the importance of structured intermediary systems, including certification programs, regulatory frameworks, and public awareness initiatives that contribute to the professionalization of inheritors and enhance their social influence. Overall, these findings suggest that while intermediary mechanisms play a positive role in both countries, they are more decisive in the Korean model, where greater systematization and professionalization have been achieved.

**4.3.3 Analysis of the relationship test results between inheritance environment and inheritance subject.** The Inheritance Environment has a significant positive effect on the Inheritance Subject in both China and South Korea ($\beta$_China = 0.307, $p < .001$; $\beta$_Korea = 0.404, $p < .001$). In China, government protection policies and the integration of folk sports with tourism and related industries enhance inheritors' willingness and capacity for participation, although the viability of traditional practices remains under pressure. In South Korea, the stronger effect reflects the establishment of a supportive ecological system, with coherent policies and social participation that provides a more stable foundation for transmission. Overall, these findings indicate that while environmental factors play an important role in both contexts, they exert a greater influence in the Korean model, where policy-driven support and cultural integration are more fully institutionalized.

**4.3.4 Analysis of the relationship test results between inheritance subject and inheritance effect.** The Inheritance Subject significantly promotes the Inheritance Effect in both China and South Korea ($\beta$_China = 0.368, $p < .001$; $\beta$_Korea = 0.515, $p < .001$). In China, inheritors play an active role in cultural transmission, with their skills and commitment supporting the expansion of the inheritance base and participant numbers, although structural constraints may constrain the full realization of their impact. In South Korea, the stronger effect highlights the advantages of systematic training and certification, which enable inheritors to transmit and actualize cultural values more effectively. Overall, while inheritors are central in both contexts, their influence is more pronounced in the Korean model, where institutional reinforcement enhances their capacity to achieve broader and more tangible outcomes.

**4.3.5 Analysis of the mediating effect of inheritance subject.** Using the bootstrap method in AMOS 26.0, this study tested whether the Inheritance Subject mediates the relationship between the Inheritance Object, Inheritance Intermediary, Inheritance Environment, and Inheritance Effect. Initially, 336 valid samples from China and 331 valid samples from South Korea were selected as the bootstrap population. Random samples were drawn, and 5,000 resamples were conducted. To assess the significance of the mediating effect, a 95% confidence interval (CI) was employed to determine whether the interval included 0. If the interval did not contain 0, the mediating effect was considered significant.

Bootstrap analysis confirms that the Inheritance Subject significantly mediates the effects of the Inheritance Object, intermediary, and environment on the Inheritance Effect in both China and South Korea (Table 7). These results indicate that external supports—whether cultural, organizational, or environmental—influence inheritance outcomes primarily through the agency and capacity of inheritors. In both contexts, inheritors serve as the pivotal link translating resources into the actualization of cultural transmission. The mediating role is stronger in South Korea, reflecting the effectiveness of its systematic training and certification in enhancing the capacity of inheritors. Overall, these findings underscore the central position of inheritors in the inheritance system, highlighting that investment in their skills and recognition is essential for sustaining folk sports.

## 5 Discussion

### 5.1 Comparative analysis of the inheritance object dimension

Our comparative analysis reveals a fundamental divergence in the role of the Inheritance Object between China and South Korea. The significantly stronger effect observed in China ($\beta$_China = 0.390; $\beta$_Korea = 0.161) underscores its role as a primary cultural anchor, where inheritance depends on the intrinsic value and historical significance of the

sports, extending Jiang's conceptualization and highlighting vulnerability to stagnation without innovation [28]. By contrast, the weaker effect in South Korea challenges the universality of this object-centric model, demonstrating how an advanced institutional ecosystem—including disciplinary education, certification, and programmatic support—reduces reliance on the object's inherent appeal [30,31]. This evidence refines existing theoretical frameworks by showing that the Inheritance Object operates not as a static driver but as a dynamic factor mediated by institutional support. It serves as a foundational motivator in China's culturally driven model and as an integrated element within Korea's institutionalized system [9,10].

### 5.2 Comparative analysis of the inheritance intermediary dimension

The disparity in the influence of inheritance intermediaries between South Korea ($β\_Korea = 0.352$) and China ($β\_China = 0.181$) indicates a critical difference in institutionalization. Korea's stronger effect derives from a structured intermediary framework that integrates formal certification, standardized regulatory protocols, and digital dissemination strategies. This framework professionalizes inheritors and enhances their societal impact, consistent with evidence on technology-mediated heritage promotion [32]. By contrast, China's weaker effect reflects a less integrated approach: traditional community-based transmission continues, but it lacks the institutional scaffolding required to achieve greater scalability and influence. These results extend the model proposed by Ma and Wang by quantitatively demonstrating that intermediary effectiveness depends not only on presence but also on systemic integration and professionalization [10]. In this sense, Korean intermediaries function as force multipliers within a coherent heritage ecosystem, whereas their Chinese counterparts serve primarily as facilitators within fragmented local contexts. This finding reframes intermediaries from passive channels as active architects of sustainability.

### 5.3 Comparative analysis of the inheritance environment dimension

The comparative results indicate that South Korea's Inheritance Environment has a stronger positive effect on the Inheritance Subject than China ($β\_Korea = 0.404$; $β\_China = 0.307$), reflecting the role of coherent policy frameworks and systematic integration. In China, abundant regional cultural resources and rapid social development coexist with uneven safeguards, rendering many projects vulnerable to decline [33]. By contrast, Korea's deliberate national strategy embeds heritage protection within broader economic and social policies, ensuring institutional coherence and long-term sustainability [34,35]. This pattern is consistent with studies emphasizing that ecological and cultural coordination is essential for sustaining traditional sports, highlighting how institutional design conditions the effectiveness of environmental inputs [36]. It also aligns with comparative appraisals of the UNESCO framework, which stresses that safeguarding intangible heritage requires structural integration rather than symbolic recognition [37]. Overall, the environment dimension should be understood not merely as a contextual backdrop but as a strategically engineered enabler that shapes the effectiveness of the inheritance system.

### 5.4 Comparative analysis of the inheritance subject dimension

The Inheritance Subject shows a substantially stronger effect on inheritance outcomes in South Korea ($β\_Korea = 0.515$; $β\_China = 0.368$), underscoring its pivotal role in translating resources into effective cultural transmission. In China, family- and community-based models cultivate deeply embodied skills but lack structured pathways for large-scale talent development. By contrast, Korea's advantage lies in a comprehensive talent development framework that integrates formal education, credentialing systems, and public engagement platforms, thereby strengthening the agency of inheritors [9,30]. Qualitative evidence further indicates that Korean sport exhibits a synergistic pattern of transmission, cultural coexistence, and innovation across folk festivals, school sports, and elite competition, which broadens participation channels and visibility [31]. This comparative pattern confirms that subject effectiveness depends not only on individual expertise

but also on the institutional conditions supporting and legitimizing inheritors [38]. Overall, the subject dimension functions as the central conduit of the inheritance system, mediating the effects of object, intermediary, and environment on cultural outcomes.

### 5.5 Comparative analysis of the inheritance effect dimension

The pathway analysis shows that the Inheritance Effect is significantly stronger in South Korea than in China, a disparity largely explained by the greater effectiveness of the Inheritance Subject ($\beta$_Korea = 0.515; $\beta$_China = 0.368). In China, although the number of inheritors is large, decentralized and fragmented practices diminish overall impact, echoing spatial distribution patterns identified by Hu et al. [39]. Our findings indicate that this stems from a reliance on individual transmission, which preserves stylistic diversity but limits scalability and coordination. By contrast, Korea's stronger outcome results from a centrally coordinated yet locally implemented model that integrates organizational structures, expands inheritance bases, and fosters public engagement [40]. This aligns with Kurin's argument that sustainable heritage requires "systems of continuance", while refining this argument by demonstrating that centralization must be paired with professionalization [37]. Conversely, the Chinese case illustrates the challenge emphasized by Cui et al., namely that even a pivotal Inheritance Subject cannot fully maximize effectiveness without robust institutional coupling [9]. The task for China is thus not merely to expand numbers but to construct integrative frameworks that transform grassroots potential into systematic and sustainable cultural transmission.

### 5.6 Comparative analysis of the mediating role of inheritance subject

The bootstrap results confirm that the Inheritance Subject plays a significant mediating role in both countries, but the strength and form of this mediation vary, reflecting contrasting institutional logics. In China, mediation occurs mainly through familial and regionally embedded mechanisms, relying on experiential knowledge and oral tradition. While this mechanism preserves localized authenticity, it lacks systematic support and constrains scalability, consistent with Huang et al.'s observation on the constraints of informal transmission networks [41]. Our model further specifies that such weak institutionalization attenuates the overall Inheritance Effect.

By contrast, mediation in South Korea is explicitly formalized and systematized: inheritors act as culturally skilled professionals who draw upon pedagogical training, technological integration, and policy-backed platforms to extend their influence [36]. This supports Kurin's argument that safeguarding depends on "professionalizing tradition", while extending it by demonstrating how professionalization enhances the mediating role of the Inheritance Subject [37]. The stronger mediating effects observed in Korea underscore that institutionalization does not bypass inheritors but enhances their agency as central conduits of cultural transmission. Thus, although both countries hinge on the Inheritance Subject, Korea's experience shows that mediation is not only a cultural process but also an institutional achievement. For China, the priority is not to replace traditional modes but to develop hybrid structures providing systematic support while maintaining grassroots legitimacy.

## 6 Conclusion and future directions

### 6.1 Research conclusion

This paper constructs a structural equation model to analyze the influencing factors of folk sports inheritance and provides a detailed comparative analysis between China and South Korea. The study reveals that, while there are differences in the influencing factors of folk sports inheritance between the two countries, there is also considerable potential for mutual learning and exchange. Both China and South Korea demonstrate positive direct effects from the Inheritance Object, Inheritance Intermediary, and Inheritance Environment on the Inheritance Subject. This suggests that, in both countries, the Inheritance Object, intermediary, and environment are key factors driving the active participation of the Inheritance

Subject. Additionally, the positive effect of the Inheritance Subject on the Inheritance Effect emphasizes the importance both countries place on the skill levels of inheritors and the organization of inheritance activities.

While similarities exist in the influencing factors of inheritance, significant differences are observed in their specific performance. South Korea excels in areas such as inheritor recognition and training, the establishment of inheritance systems, and the creation of inheritance ecological environments, giving it a relative advantage in overall performance. In contrast, although Chinese folk sports inheritors excel in skill level, there are gaps in inheritance awareness, inheritor training, and the protection of the inheritance ecological environment.

## 6.2 Theoretical contributions

This study makes several important theoretical contributions:

Innovative Theoretical Framework: This study develops an innovative theoretical framework by systematizing the influencing factors of folk sports inheritance into five dimensions: Inheritance Subject, Inheritance Object, Inheritance Intermediary, Inheritance Environment, and Inheritance Effect. This framework creates a new, comprehensive evaluation model that not only organizes the influencing factors systematically but also validates the scientific and practical nature of the model through a comparative empirical study between China and South Korea.

Enhancing Cross-Cultural Studies: Through a comparative analysis of China and South Korea, this study enriches cross-cultural research on folk sports inheritance. It deepens our understanding of how folk sports are inherited in different cultural contexts, revealing both universal patterns and context-specific factors. This offers new theoretical insights for international comparative studies on folk sports inheritance.

Central Role of the Inheritance Subject: The study highlights the central role of the Inheritance Subject in the transmission process. It finds that the Inheritance Object, Intermediary, and Environment all influence the Inheritance Effect through the Inheritance Subject. This clarifies the mediating role of the Inheritance Subject within the overall system and expands the research on the mechanisms behind folk sports inheritance. This finding aligns with Kurin's observations in American folk culture preservation, reinforcing the cross-cultural validity of the Inheritance Subject's mediating function [37].

## 6.3 Policy contributions

In addition to its theoretical innovations, this study provides valuable policy implications for cultural heritage governance at both the national and bilateral levels.

First, the five-dimensional model developed in this research offers a practical diagnostic tool for cultural institutions to evaluate their folk sports transmission systems. By identifying structural imbalances—such as an overemphasis on material documentation or underdeveloped intermediary mechanisms—this model can inform more targeted and holistic strategies in policy formulation.

Second, the model provides a basis for improving institutional practices. It suggests that cultural heritage protection should go beyond object-centered approaches and instead invest in cultivating local leadership (Inheritance Subject), building community-based transmission platforms (Inheritance Intermediary), and creating more integrated policy environments (Inheritance Environment). These insights can guide cultural bureaus, heritage centers, and educational institutions in enhancing the sustainability of folk sports practices.

Third, the model opens up new pathways for bilateral cooperation between China and South Korea. Shared inventories of folk sports heritage, joint training programs for youth practitioners, and cross-border evaluation standards based on the five dimensions can foster mutual learning, strategic alignment, and long-term cultural exchange.

## 6.4 Recommendations for improving folk sports inheritance in China

Based on the research findings, the inheritance of folk sports in China can be improved in several key areas:

Systematic Training Mechanism for Inheritance Subjects: Establish a more structured and professional training system for Inheritance Subjects to improve their educational and teaching capabilities, as well as organizational development skills. This includes developing clear criteria for Inheritance Subject recognition, creating a tiered training system, implementing regular skill training and instructional guidance, and introducing a "master-apprentice" mentorship model. These efforts will form a well-organized team of Inheritance Subjects, enhancing their overall effectiveness in the inheritance process.

Legal Protection and Regulatory Framework: Formulate and improve relevant laws and regulations to provide legal protection for the inheritance of folk sports. Drawing from South Korea's experience, China could establish specialized protection laws for folk sports, defining the rights and responsibilities of the government, social organizations, and individuals involved. This could include setting up a project evaluation mechanism and creating dedicated funding sources for folk sports.

Creating an Ecological Environment for Inheritance: Collaborate between government and society to create a supportive ecological environment for folk sports inheritance. The government should increase policy support and financial investment to build dedicated inheritance bases. Social organizations and enterprises can also contribute through sponsorships, venue support, and organizing events, while encouraging grassroots initiatives to engage in inheritance activities.

Integration with Other Industries: Strengthen the integration of folk sports with related industries, such as tourism and education, to expand inheritance channels. Incorporating folk sports into school curricula, developing specialized tourism projects, and organizing festivals and sporting events will provide new opportunities for cultural transmission. By innovating inheritance methods through industry integration, the influence and economic benefits of folk sports can be enhanced, ensuring their sustainable development.

## 6.5 Recommendations for improving folk sports inheritance in South Korea

Although South Korea has established a relatively mature system for folk sports inheritance, there remains room for innovation and enhancement in the context of globalization, technological advancement, and intergenerational change. Based on the findings of this study, several targeted recommendations are proposed:

First, establish a dynamic assessment mechanism to identify and prioritize marginalized, endangered, or region-specific folk sports not yet included in the national intangible cultural heritage list. This should include regular evaluation protocols and differentiated resource allocation based on urgency levels and regional significance.

Second, leverage South Korea's technological expertise to create innovative digital preservation and transmission platforms. This includes developing VR-based experiential learning environments, AI-assisted cultural documentation systems, and interactive mobile applications that can engage younger demographics while maintaining cultural authenticity.

Third, transition from top-down institutional preservation toward community-driven revitalization models. This involves establishing participatory governance structures that empower local communities, grassroots organizations, and youth groups to take ownership of folk sports transmission, ensuring cultural practices remain living traditions rather than static exhibits.

Finally, capitalize on South Korea's global cultural influence through strategic integration of folk sports into existing cultural export channels. This includes embedding traditional sports elements into K-culture content, establishing international folk sports festivals, and developing educational exchange programs that position South Korea as a leader in innovative cultural heritage preservation.

## 6.6 Future directions

This study establishes a theoretical foundation for understanding the mechanisms underlying the inheritance of folk sports in both China and South Korea and provides a reference framework that may inform similar studies in other countries

and regions. However, it is important to acknowledge that the five-dimensional model proposed herein—comprising the Inheritance Subject, Inheritance Object, Inheritance Intermediary, Inheritance Environment, and Inheritance Effect—was developed within the socio-cultural contexts of East Asia. In particular, the model reflects characteristics of Confucian-influenced societies, including a strong tradition of collectivism, hierarchical social structures, and government-led cultural preservation. These cultural attributes may not be equally salient in other regions, thereby limiting the model's generalizability and direct applicability.

Future research should thus refrain from assuming the universality of this framework and instead empirically test its validity across diverse cultural contexts. Comparative studies involving countries with distinct historical trajectories, legal frameworks, and sports governance systems—such as those in Europe, Africa, or Latin America—are necessary to assess the model's adaptability and potential for refinement. Such investigations may uncover culturally specific variables, reveal implicit assumptions embedded in the current framework, and contribute to the development of a more nuanced and inclusive model suitable for broader cross-cultural application.

Furthermore, scholars should remain critically aware of the risks associated with cultural essentialism and ethnocentric bias in cross-national research. Attention should be paid to the practical realities and localized cultural traditions of each society when designing context-appropriate inheritance strategies. In an increasingly globalized world, future studies should also explore how transnational cultural flows, digital media platforms, and evolving policy landscapes influence the transmission of folk sports. Such factors may not only reshape the modalities of cultural inheritance but also alter the meaning, function, and sustainability of folk sports traditions in contemporary societies.

## Supporting information

**S1 File. Sample Data-China.**
(XLSX)

**S2 File. Sample Data–South Korea.**
(XLSX)

## Author contributions

**Conceptualization:** Xiaobin Jin, Yu Hou.

**Data curation:** Xiaobin Jin, Yu Hou.

**Formal analysis:** Xiaobin Jin, Yu Hou.

**Funding acquisition:** Xiaobin Jin, Yu Hou.

**Investigation:** Xiaobin Jin, Yu Hou.

**Methodology:** Xiaobin Jin, Yu Hou.

**Project administration:** Xiaobin Jin, Yu Hou.

**Resources:** Xiaobin Jin, Yu Hou.

**Software:** Xiaobin Jin, Yu Hou.

**Supervision:** Xiaobin Jin, Yu Hou.

**Validation:** Xiaobin Jin, Yu Hou.

**Visualization:** Yu Hou.

**Writing – original draft:** Xiaobin Jin, Yu Hou.

**Writing – review & editing:** Xiaobin Jin, Yu Hou.

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
