## [Decision Letter · Decision Letter 0]

8 Jul 2025

Dear Dr. Hou,

Thank you for submitting your manuscript to PLOS ONE. After careful consideration, we feel that it has merit but does not fully meet PLOS ONE’s publication criteria as it currently stands. Therefore, we invite you to submit a revised version of the manuscript that addresses the points raised during the review process.

We look forward to receiving your revised manuscript.

Kind regards,

Assoc. Prof. Phakkharawat Sittiprapaporn, Ph.D.

Academic Editor

PLOS ONE

Journal Requirements:

Reviewers' comments:

Reviewer's Responses to Questions

**Comments to the Author**

1. Is the manuscript technically sound, and do the data support the conclusions?

Reviewer #1: Partly

Reviewer #2: Partly

2. Has the statistical analysis been performed appropriately and rigorously?

Reviewer #1: Yes

Reviewer #2: Yes

3. Have the authors made all data underlying the findings in their manuscript fully available?

Reviewer #1: Yes

Reviewer #2: No

4. Is the manuscript presented in an intelligible fashion and written in standard English?

Reviewer #1: Yes

Reviewer #2: No

Reviewer #1: I have taken note of the manuscript submitted to me for review, entitled “A Comparative Study on Folk Sports Inheritance in China and South Korea: Influencing Factors and Mediating Roles”.

To improve the quality of the article published in this prestigious journal, I make some observations below that are detailed and can be corrected by the authors.

- Although relevant literature exists, the review could delve more deeply into the theoretical framework of intangible cultural heritage from anthropology or cultural studies. In this way, the research topic could be better understood.

- In the research design, it is more pertinent to put the sample or participants first to situate the reader the sample under study.

- p" in the level of significance is in italics, both at the foot of the tables and in the text.

- Statistical values should be without 0 before the comma, e.g., ** p < .01 Same in tables and text.

- Broader structural or historical factors that might explain the observed differences are not sufficiently discussed. For example, the text between lines 419 and 427, between 462 and 482 or between 485 and 509, do not have any citations to support the findings obtained, but are expressions to justify and argue the data. Therefore, they must rely on existing literature to contrast the empirical evidence obtained with previous studies.

- It is implicitly assumed that the proposed model is universal, without discussing its cultural limitations or its transferability to other contexts.

- The recommendations made are mostly one-sided (towards China) and no innovative ideas are offered for South Korea.

- I invite the authors to use other citations to support the statements made in the text, since their bibliographic references, in their entirety, are centered on authors from the Asian continent. In this way, it is possible to enrich the perspective and the variables under study of other citizens at the international level.

Reviewer #2: Thank you for the opportunity to review your manuscript, “A Comparative Study on Folk Sports Inheritance in China and South Korea.” Your work addresses an important and under-explored area by offering a comparative analysis of folk sports heritage systems in two distinct cultural contexts.

However, several revisions are required before the manuscript can be considered suitable for publication:

Theoretical Clarity: The conceptual framework requires stronger grounding in relevant literature. Please clarify how the five dimensions relate to existing theories of cultural transmission, sports sociology, or heritage studies.

Statistical Reporting: While the use of SEM is appropriate, key statistical details—such as model fit indices and assumption testing—are missing. Please report these clearly and transparently.

Data Availability: In line with journal policy, the dataset underlying your analysis must be made fully accessible, either through a public repository or supplementary files.

Language and Presentation: The manuscript contains numerous grammatical, structural, and stylistic issues. We strongly recommend professional English language editing to ensure clarity and consistency.

Policy Implications: The practical contribution of your findings could be strengthened. Consider elaborating on how your model informs institutional practice or bilateral collaboration in cultural preservation.

**Do you want your identity to be public for this peer review?** For information about this choice, including consent withdrawal, please see our Privacy Policy

Reviewer #1: **Yes: ** Miguel Ángel Durán-Vinagre

Reviewer #2: No

---

## [Author Response · Author response to Decision Letter 1]

1 Aug 2025

Reviewer #1

1.The theoretical foundation of the research on the influence of folk sports has been further enriched from multidisciplinary perspectives such as anthropology, cultural studies, management, and intangible cultural heritage studies. The additional content can be found in lines 151-168 of the manuscript.

2.The research sample has been placed at the beginning of the Research Design section, with corresponding adjustments made to other content.

3.We appreciate the reviewer for pointing out this issue. The revisions have been made.

4.We appreciate the reviewer for pointing out this issue. The revisions have been made.

5.References have been added to strengthen the argument and provide a comparison with previous research.

6.The "Future Directions" section of the manuscript has been adjusted and expanded, with a discussion on its cultural limitations.

7.A new section, "6.5 Recommendations for Improving Folk Sports Inheritance in South Korea," has been added to address and correct the limitation of previously providing recommendations solely for China.

8.The sources of the references have been diversified, with the addition of references that offer a broader international perspective.

Reviewer #2

1.The theoretical foundation of the research on the influence of folk sports has been further enriched from multidisciplinary perspectives such as anthropology, cultural studies, management, and intangible cultural heritage studies. The additional content can be found in lines 151-168 of the manuscript.

2.The author has re-examined the manuscript and confirmed that it presents the key model fit indices and assumption testing.

3.The original data have been uploaded to the Supplementary Files.

4.Thank you for your valuable feedback. We apologize for the language issues in our manuscript. We have worked on the manuscript for a long time, and the repeated addition and removal of sentences and sections has unfortunately impacted its readability. We have now made revisions to both the language and readability, and have involved native English speakers for language corrections. We sincerely hope that the flow and language quality have been significantly improved.

5.A new section, "Policy Contributions," has been added in Section 6.3, where we systematically elaborate on how the five-dimensional model proposed in this study can provide evaluation and intervention pathways for cultural institutions. Additionally, the specific applications of this model in the collaboration of folk sports between China and South Korea are discussed.

---

## [Decision Letter · Decision Letter 1]

19 Aug 2025

Dear Dr. Hou,

Thank you for submitting your manuscript to PLOS ONE. After careful consideration, we feel that it has merit but does not fully meet PLOS ONE’s publication criteria as it currently stands. Therefore, we invite you to submit a revised version of the manuscript that addresses the points raised during the review process.

We look forward to receiving your revised manuscript.

Kind regards,

Asoc. Prof. Phakkharawat Sittiprapaporn, Ph.D.

Academic Editor

PLOS ONE

Journal Requirements:

Reviewers' comments:

Reviewer's Responses to Questions

**Comments to the Author**

Reviewer #1: All comments have been addressed

Reviewer #2: (No Response)

2. Is the manuscript technically sound, and do the data support the conclusions?

Reviewer #1: Yes

Reviewer #2: (No Response)

3. Has the statistical analysis been performed appropriately and rigorously?

Reviewer #1: Yes

Reviewer #2: (No Response)

4. Have the authors made all data underlying the findings in their manuscript fully available?

Reviewer #1: Yes

Reviewer #2: (No Response)

5. Is the manuscript presented in an intelligible fashion and written in standard English?

Reviewer #1: Yes

Reviewer #2: (No Response)

Reviewer #1: The authors have addressed the comments and suggestions for improvements made in the previous reviews by the various reviewers. This improves the quality of the manuscript and complies with the guidelines and ethics of the prestigious PlosOne journal.

Reviewer #2: This manuscript is a re-submission following earlier reviewer feedback. The revisions have addressed several prior concerns, including improved figure captions and expanded engagement with recent literature. The topic remains relevant and timely, and the empirical approach is appropriate for the aims of the study. However, despite these improvements, there are still important areas that require further work to bring the paper to PLOS ONE’s publication standard. Specifically, the framing of the research gap, methodological transparency, and depth of discussion continue to limit the manuscript’s scholarly impact.

Here are my specific concerns:

1. While the introduction has been refined since the initial submission, it still lacks a sharply defined research gap and explicit statement of novelty relative to prior studies.

2. The link between the literature review and the specific aims of this paper is not as strong as it could be, especially in justifying why this study is needed now.

3. Although some methodological clarifications were made since the last submission, key details are still missing, including sampling procedures, justification for variable inclusion/exclusion, and handling of missing data.

4. While results tables and figures are now clearer, effect sizes are still inconsistently reported alongside p-values.

5. The narrative continues to repeat table content rather than interpret it.

6. Although expanded, the discussion still tends to summarise rather than critically engage with the literature. Also, over-generalisation remains a concern; conclusions should be more closely tied to the scope of the sample and study design.

**Do you want your identity to be public for this peer review?** For information about this choice, including consent withdrawal, please see our Privacy Policy

Reviewer #1: **Yes: ** Miguel Ángel Durán-Vinagre

Reviewer #2: **Yes: ** Richard Peter Bailey

---

## [Author Response · Author response to Decision Letter 2]

8 Sep 2025

Response to Reviewer #1:

We sincerely thank Reviewer #1 for the positive evaluation of our revised manuscript. We greatly appreciate your recognition of the improvements made in response to previous comments and suggestions. Your feedback confirms that the revisions have enhanced the quality of the manuscript and ensured compliance with the guidelines and ethical standards of PLOS ONE.

Response to Reviewer #2:

We sincerely thank Reviewer #2 for the detailed and constructive feedback on our revised manuscript. We greatly appreciate your recognition of the improvements made in figure captions, literature engagement, and the overall empirical approach. Below, we provide point-by-point responses to your specific concerns.

Comment 1: While the introduction has been refined since the initial submission, it still lacks a sharply defined research gap and explicit statement of novelty relative to prior studies.

Response 1:

In the revised Introduction (lines 60–69, 93–103), we explicitly addressed this concern by: (1) identifying three specific research gaps—namely, the predominant reliance on qualitative methodologies, the scarcity of cross-national comparative studies, and the absence of a unified analytical framework; (2) linking each gap directly to the corresponding contributions of this study; and (3) highlighting the novelty of introducing and empirically validating a five-dimensional framework using cross-national data. These revisions aim to clearly delineate the study’s rationale and scholarly contribution.

Comment 2: The link between the literature review and the specific aims of this paper is not as strong as it could be, especially in justifying why this study is needed now.

Response 2:

In the revised manuscript (lines 174–186), we strengthened the link between the literature review and the research aims by emphasizing the timeliness of the study. Specifically, China’s rapid modernization is accelerating the decline of traditional folk sports, whereas Korea’s institutionalized preservation through UNESCO recognition and legal frameworks provides a mature model. This contrast highlights the urgency of conducting a cross-national comparison. Moreover, the identified gaps—namely, the reliance on qualitative studies, the absence of cross-national frameworks, and the lack of unified evaluation indices—are now directly linked to our proposed quantitative, five-dimensional approach, clarifying the necessity and relevance of the present study.

Comment 3: Although some methodological clarifications were made since the last submission, key details are still missing, including sampling procedures, justification for variable inclusion/exclusion, and handling of missing data.

Response 3:

In the revised manuscript (lines 222–233, Section 3.1: Research Sample), we thank the reviewer for emphasizing the need for methodological clarity. We have added a detailed description of our stratified sampling procedure across stakeholder groups in both countries, as well as the rationale for inclusion and exclusion of variables based on prior theoretical models and pilot testing. Additionally, all questionnaires were fully completed, resulting in no missing data, and thus no imputation procedures were required. These revisions enhance transparency and ensure methodological rigor, consistent with best practices for global and cross-cultural research.

Comment 4: While results tables and figures are now clearer, effect sizes are still inconsistently reported alongside p-values.

Response 4:

In the revised manuscript (lines 415–422, Section 4.3: Empirical Results), we updated Tables 6 and 7 to consistently report standardized effect sizes (β coefficients) alongside p-values and their 95% confidence intervals. For mediation analyses, standardized indirect effects and bias-corrected bootstrap confidence intervals were also included. These revisions improve the interpretability of our results by balancing statistical significance with substantive effect size and ensure alignment with standard reporting practices in structural equation modeling (SEM).

Comment 5: The narrative continues to repeat table content rather than interpret it.

Response 5:

In the revised manuscript (lines 431–479, 489–498), Sections 4.3.1–4.3.5 were carefully revised to minimize redundant repetition of tabulated data and to strengthen interpretative commentary. Rather than restating coefficients and test statistics already presented in the tables, the text emphasizes observed differences between China and South Korea, highlights underlying mechanisms (e.g., cultural identity versus institutional reinforcement), and draws comparative insights. Each subsection concludes with a concise synthesis that underscores key implications while avoiding over-generalization.

Comment 6: Although expanded, the discussion still tends to summarise rather than critically engage with the literature. Also, over-generalisation remains a concern; conclusions should be more closely tied to the scope of the sample and study design.

Response 6:

In the revised manuscript (lines 503–606, Sections 5.1–5.6), we substantially refined the discussion to enhance interpretative depth and scholarly engagement. Rather than reiterating tabulated results, each subsection emphasizes explanatory interpretation, cross-national contrasts, and theoretical implications. For instance, the role of the Inheritance Object as a cultural anchor in China versus an institutional component in Korea is highlighted in relation to Jiang[30] and Kurin[39]. Recent studies (e.g., Hong et al.[34]; Chen & Liu[36]; Sun[38]) are integrated to critically examine how intermediaries and environments function as multiplicative factors within institutional systems. We explicitly connect effect size differences (e.g., β values for China vs. Korea) to systemic contrasts, while engaging with comparative frameworks from Cui et al.[9] and Koo[42]. These revisions reduce redundancy, strengthen scholarly dialogue, and more firmly situate our findings within existing debates, providing a clearer and more focused presentation of the study’s contributions.

---

## [Decision Letter · Decision Letter 2]

30 Oct 2025

A Comparative Study on Folk Sports Inheritance in China and South Korea: Influencing Factors and Mediating Roles

PONE-D-25-25970R2

Dear Dr. Hou,

We’re pleased to inform you that your manuscript has been judged scientifically suitable for publication and will be formally accepted for publication once it meets all outstanding technical requirements.

Kind regards,

Assoc. Prof. Dr. Phakkharawat Sittiprapaporn, Ph.D.

Academic Editor

PLOS ONE

Additional Editor Comments (optional):

Reviewers' comments:

Reviewer's Responses to Questions

**Comments to the Author**

Reviewer #2: All comments have been addressed

2. Is the manuscript technically sound, and do the data support the conclusions?

Reviewer #2: Yes

3. Has the statistical analysis been performed appropriately and rigorously?

Reviewer #2: Yes

4. Have the authors made all data underlying the findings in their manuscript fully available?

Reviewer #2: Yes

5. Is the manuscript presented in an intelligible fashion and written in standard English?

Reviewer #2: Yes

Reviewer #2: The revised manuscript, A Comparative Study on Folk Sports Inheritance in China and South Korea: Influencing Factors and Mediating Roles, reflects a thorough and effective response to all reviewer feedback. The research gap and contribution are now clearly articulated, with improved methodological transparency through detailed sampling procedures and justification of variables. Statistical reporting has been strengthened by the inclusion of standardised effect sizes and confidence intervals. The discussion section has been substantially reworked to offer critical interpretation and meaningful engagement with relevant literature, moving beyond descriptive summary. Ethical approval, data availability, and adherence to journal standards are clearly documented. Overall, the revision demonstrates significant improvement in coherence, analytical rigour, and scholarly merit. The manuscript is suitable for publication, subject to minor editorial refinements in language and style.

**Do you want your identity to be public for this peer review?** For information about this choice, including consent withdrawal, please see our Privacy Policy

Reviewer #2: **Yes: ** Richard Peter Bailey

---

## [Editor Report · Acceptance letter]

PONE-D-25-25970R2

PLOS ONE

Dear Dr. Hou,

I'm pleased to inform you that your manuscript has been deemed suitable for publication in PLOS ONE. Congratulations! Your manuscript is now being handed over to our production team.

Kind regards,

on behalf of

Assoc. Prof. Dr. Phakkharawat Sittiprapaporn

Academic Editor

PLOS ONE